# Phytocannabinoids: Chromatographic Screening of Cannabinoids and Loading into Lipid Nanoparticles

**DOI:** 10.3390/molecules28062875

**Published:** 2023-03-22

**Authors:** Aleksandra Zielińska, Raquel da Ana, Joel Fonseca, Milena Szalata, Karolina Wielgus, Faezeh Fathi, M. Beatriz P. P. Oliveira, Rafał Staszewski, Jacek Karczewski, Eliana B. Souto

**Affiliations:** 1Institute of Human Genetics, Polish Academy of Sciences, Strzeszyńska 32, 60-479 Poznan, Poland; 2Department of Pharmaceutical Technology, Faculty of Pharmacy, University of Porto, Rua de Jorge Viterbo Ferreira, 228, 4050-313 Porto, Portugal; 3Department of Biotechnology, Institute of Natural Fibres and Medicinal Plants, National Research Institute, Wojska Polskiego 71B, 60-630 Poznan, Poland; 4Department of Pediatric Gastroenterology and Metabolic Diseases, Poznan University of Medical Sciences, Szpitalna 27/33, 60-572 Poznan, Poland; 5REQUIMTE/LAQV, Department of Chemical Sciences, Faculty of Pharmacy, University of Porto, Rua Jorge Viterbo Ferreira No. 280, 4050-313 Porto, Portugal; 6Department of Hypertension Angiology and Internal Medicine, Poznan University of Medical Sciences, 61-701 Poznan, Poland; 7Department of Environmental Medicine, Poznan University of Medical Sciences, 61-701 Poznan, Poland; 8Department of Gastroenterology, Dietetics and Internal Diseases, H. Swiecicki University Hospital, Poznan University of Medical Sciences, 60-355 Poznan, Poland; 9REQUIMTE/UCIBIO, Faculty of Pharmacy, University of Porto, Rua de Jorge Viterbo Ferreira, 228, 4050-313 Porto, Portugal; 10Associate Laboratory i4HB—Institute for Health and Bioeconomy, Faculty of Pharmacy, University of Porto, 4050-313 Porto, Portugal

**Keywords:** solid lipid nanoparticles, nanostructured lipid carriers, cannabidiol, viscoelastic behavior, Compritol^®^ 888 ATO, Miglyol^®^ 812

## Abstract

Solid Lipid Nanoparticles (SLN) and Nanostructured Lipid Carriers (NLC) are receiving increasing interest as an approach to encapsulate natural extracts to increase the physicochemical stability of bioactives. Cannabis extract-derived cannabidiol (CBD) has potent therapeutic properties, including anti-inflammatory, antioxidant, and neuroprotective properties. In this work, physicochemical characterization was carried out after producing Compritol-based nanoparticles (cSLN or cNLC) loaded with CBD. Then, the determination of the encapsulation efficiency (EE), loading capacity (LC), particle size (Z-Ave), polydispersity index (PDI), and zeta potential were performed. Additionally, the viscoelastic profiles and differential scanning calorimetry (DSC) patterns were recorded. As a result, CBD-loaded SLN showed a mean particle size of 217.2 ± 6.5 nm, PDI of 0.273 ± 0.023, and EE of about 74%, while CBD-loaded NLC showed Z-Ave of 158.3 ± 6.6 nm, PDI of 0.325 ± 0.016, and EE of about 70%. The rheological analysis showed that the loss modulus for both lipid nanoparticle formulations was higher than the storage modulus over the applied frequency range of 10 Hz, demonstrating that they are more elastic than viscous. The crystallinity profiles of both CBD-cSLN (90.41%) and CBD-cNLC (40.18%) were determined. It may justify the obtained encapsulation parameters while corroborating the liquid-like character demonstrated in the rheological analysis. Scanning electron microscopy (SEM) study confirmed the morphology and shape of the developed nanoparticles. The work has proven that the solid nature and morphology of cSLN/cNLC strengthen these particles’ potential to modify the CBD delivery profile for several biomedical applications.

## 1. Introduction

Lipid nanoparticles are one of the most popular drug delivery systems due to their versatility in loading chemically different active ingredients with high efficiency [1]. The primary purpose of using these particles is to increase the bioavailability of the loaded drug [2]. There are two types of lipid nanoparticles, namely Solid Lipid Nanoparticles (SLN) and Nanostructured Lipid Carriers (NLC), which are composed exclusively of solid oils and a mixture of solid and liquid oils, respectively [3,4]. Both generations of lipid nanoparticles are composed of two immiscible phases: lipid and water. The main difference in the structure of both nanostructures is the different distribution of the active substance in the space of the lipid matrix. Therefore, their synthesis requires surfactants. The most important properties of lipid nanoparticles include adhesion properties leading to the formation of a hydrophobic film on the skin, which limits the transepidermal water loss and the modified release of drugs [5]. However, the most significant advantages of these drug carriers are their non-toxicity, which results from a biodegradable lipid matrix that undergoes enzymatic degradation to compounds naturally present in the human body [6,7]. SLN and NLC are also relatively easy to obtain and introduce to the market due to the high availability of lipids. The potential toxicity of lipid nanoparticles may depend only on the particle size, the incorporated active substance, and the type of surfactant. Particles exceeding 40 nm cannot penetrate the living cells of the epidermis or the bloodstream and thus do not cause side effects for the body. Only compounds approved by the European Medicines Agency (EMA) and the American Food and Drug Administration (FDA) are on the GRAS list (Generally Recognized As Safe—considered safe).

Lipid nanoparticles are especially interesting for oral administration due to different aspects. Firstly, it should emphasize the mucoadhesive properties that they present due to their colloidal structure, which facilitates drug release in the intestine [8,9,10,11,12,13,14]. Additionally, they could be absorbed by the intestinal cells since the lipids composing the nanoparticles have a promoting absorption effect. Many studies have produced lipid nanoparticles to increase the limited oral bioavailability of natural extracts and protect the bioactive compounds loaded from the harsh gastric environments and hepatic first-pass metabolism [6,15,16,17]. The delayed drug release obtained by SLN and NLC enables the intact presence of the drug in the intestine [18]. Thus, some of the features that make lipid nanoparticles up-and-coming systems for oral administration are providing controlled drug release, protecting drugs from degradation, and improving oral bioavailability of the drug by reducing hepatic first-pass metabolism [19,20,21].

Moreover, the added value of SLN and NLC encapsulating natural extracts remains in their lipid composition [22,23]. The absorption-promoting effect of lipids has been studied widely, describing the degradation of the lipids by the digestive enzymes resulting in mono and diglycerides that form micelles. In these micelles, the bioactive compounds will be included and interact with the bile salts, resulting in mixed micelles that facilitate the bioactive compound absorption. Indeed, intestinal lymphatic vessels are specialized to assimilate dietary fats as long-chain fatty acids or triglycerides. This feature makes lipid nanoparticles an exciting system for drug targeting [24]. It is shown that triglyceride-based nanoparticles and long-chain rich acids-based nanoparticles may enter the intestinal lymphatic system of the gut [7,22,25]. In this manner, NLC and SLN formulations, including lipids with long-chain fatty acids (containing 12 or more carbon atoms) as Compritol^®^ 888 ATO, which consists of mono-, di- and tri-esters of behenic acid (C22), are described as sound systems to enhance natural extracts’ bioavailability [26]. Encapsulation efficiency, smaller particle size, narrow size distribution, and rheological behavior are crucial factors in obtaining efficient SLN and NLC formulations [27,28,29,30].

On the other hand, the therapeutic potential of hemp extracts, including phytocannabinoids, has been the interest of multiple scientists worldwide. *Cannabis sativa* L. is widely recognized as a non-psychotropic constituent with many medical properties for different human diseases. Cannabis-derived cannabidiol (CBD, C_21_H_30_O_2_) belongs to the group of terpenophenols. CBD has potent therapeutic properties (antioxidant, anti-inflammatory, neuroprotective) due to its already confirmed efficiency against neurological diseases or various cancer types, including neoplasms of the neural system [31,32].

The study aimed to load the selected cannabis extract (Figure 1) in both lipid carriers, namely SLN and NLC, assess its effect on the particle size, and determine the encapsulation parameters. The viscoelastic behavior and crystallinity index of the developed particles were also characterized.

## 2. Results and Discussion

### 2.1. Chromatography Analysis

The analysis of the hemp extract was performed by the High-Performance Liquid Chromatographic (HPLC), which is a validated technique that enabled quantification of 11 cannabinoids in the *Cannabis Sativa* L. extract using a wavelength of 230 nm within 22 min. All analyzed samples belonging to the same variety of *Cannabis sativa* L. did not show significant differences in the concentration of cannabinoids, as shown in Table 1. Slight discrepancies between the analyzed samples were observed, mainly in the content of CBDA (about 7%) and CBD (0.22%). The remaining cannabinoids (CBDV, CBGA, CBG, CBN, Δ^9^-THC, CBNA, CBC, THCA, and CBCA) had similar concentrations in both samples. Δ^9^-THC was determined at 0.25%, which is the limit of the permissible concentration value. According to the current regulations on the cultivation of *Cannabis sativa* L. [31,33], the total THC content cannot be higher than 0.2%. The maximum permitted limit for some cannabis varieties should be 0.6% Δ^9^-THC [31].

The results confirmed that the most abundant ingredient was CBDA, followed by CBD. The remaining compounds were present in minimal amounts, ranging from 0.04 to 0.39%. CBDA is found in raw plant matter and is an acidic precursor to active CBD, which is considered the most promising cannabinoid for its properties and applications in medicine. In addition, it is worth underlining that CBGA is a compound from which all other cannabinoids can be biosynthesized [33], the reason for its low concentration detected in both tested samples (0.04–0.09%).

### 2.2. Determination of Particle Size, Polydispersity Index, and Zeta Potential

Once the lipid nanoparticles (SLN and NLC) loaded with CBD were produced, the mean particle size (Z-Ave) and polydispersity index (PDI) of all developed formulations were determined directly after production and up to 30 days. Results on the day of production (day 0) were obtained for the empty-SLN and empty-NLC (both considered as standard samples) and reached about 201.5 ± 3.80 nm with PDI 0.281 ± 0.01 [n.u.] and 173.3 ± 4.40 nm with PDI 0.235 ± 0.02 [n.u.], respectively.

CBD-cSLN suspensions, containing 4% (*w*/*w*) of solid lipid (Compritol^®^ 888 ATO), 1.5% (*w*/*w*) of surfactant (Poloxamer^®^ 188) and 1% (*w*/*w*) of active compound (CBD extract), were produced using a probe sonication and stored at room temperature (25 °C) for 30 days. Then, Z-Ave, PDI, and zeta potential (ZP) were measured 1, 14, and 30 days after production. The results are presented in Figure 2. As a result, the Z-Ave for CBD-cSLN suspensions varied from 217.2 ± 6.50 nm (day 1) with a PDI of 0.273 ± 0.02 [n.u.]. Furthermore, after two weeks of storage at 25 °C, the particle size of cSLN suspensions loaded with CBD extract increased to 227.7 ± 11.10 nm (day 14) with a PDI of 0.301 ± 0.02 [n.u.]. It can indicate a slight agglomeration process. Finally, four weeks after the production, the CBD-cSLN suspensions gained 222 ± 5.00 nm (day 30) with a PDI of 0.320 ± 0.01 [n.u.].

Zeta potential (ZP) of empty cSLN was −18.09 ± 0.81 [mV], whereas of CBD-cSLN was −12.99 ± 0.90 [mV] (day 0) to −12.42 ± 2.14 [mV] (day 30). All of the results are presented in Table 2. It may indicate that the non-ionic nature of the surfactant has formed a spherically stabilizing adsorbed polymer layer in the SLN surface [34].

CBD-cNLC suspensions, consisting of 3% (*w*/*w*) of solid lipid (Compritol^®^ 888 ATO), 1% (*w*/*w*) of liquid lipid (Miglyol^®^ 812), 1.5% (*w*/*w*) of surfactant (Poloxamer^®^ 188) and 1% (*w*/*w*) of active compound (CBD extract), were also produced using a probe sonication and stored at room temperature for 30 days. The Z-Ave and PDI were measured 1, 14, and 30 days after production. Figure 3 shows that the Z-Ave for CBD-cSLN suspensions varied from 158.3 ± 6.60 nm (day 1), with a PDI of 0.325 ± 0.02 [n.u.]. Two weeks later, the particle size of cSLN suspensions loaded with CBD extract stored at 25 °C decreased to 128.6 ± 3.10 nm (day 14) with a PDI of 0.327 ± 0.01 [n.u.]. After one month after the production, the CBD-cSLN suspensions amounted to 146.2 ± 2.10 nm (day 30) with a PDI of 0.340 ± 0.01 [n.u.].

Zeta potential (ZP) of empty cNLC was recorded at −13.63 ± 0.89 [mV], whereas of CBD-cNLC was recorded at −5.51 ± 1.58 mV (day 0) to −12.06 ± 1.50 mV (day 30). Results are also presented in Table 3. Similar to empty-cSLN and CBD-cSLN, empty-cNLC and CBD-cNLC also showed that the presence of active compounds decreased the stability of the samples.

Zeta potential is a physical property exhibited by any particle in suspension, macromolecule, or material surface. A significant positive or negative value of the zeta potential of nanoparticles indicates good physical stability due to electrostatic repulsion. The dissociation of acidic groups on the surface of a particle will give rise to a negatively charged surface. Conversely, a primary surface will take on a positive charge. In both cases, the magnitude of the surface charge depends on the acidic or basic strengths of the surface groups and the pH of the solution. The surface charge can be reduced to zero by suppressing the surface ionization by decreasing the pH of negatively charged particles or increasing the pH of positively charged particles. The general dividing line between stable and unstable suspensions is generally taken at either +30 or −30 mV. Particles with zeta potentials more positive than +30 mV or more negative than −30 mV usually are considered stable. In our present case, our particles were slightly negative, showing that it is not the surface electrical charge that contributes to the physical stability of the particles, but rather the stereochemical hindrance promoted by the surfactants surrounding the particles.

The final size may depend on various factors, such as the chemical structure of the lipids and surfactants composing the systems and their chemical interaction. It should be pointed out that the largest particle size on day 14 corresponded to CBD-cSLN containing Compritol^®^ 888 ATO as solid lipid. NLC suspensions either consisted of one solid lipid or liquid lipid and showed significantly smaller particle size than cSLN. This could be attributed to the structural differences between lipid nanoparticles and the inclusion of Miglyol^®^ 812 in the cNLC formulations. In addition to that, PDI was similar in all CBD-loaded nanoparticles and varied in the optimal range from 0.2 to 0.3 [n.u.]. The stereochemical stability of lipid nanoparticles was monitored for 30 days. Additionally, previous research has revealed that the concentration of Miglyol^®^ 812 remarkably impacts the nanoemulsions’ stability, which is a crucial factor influencing particle size [35]. The slight reduction in particle size with the loading of CBD was attributed to the bonds between the CBD molecule and solid and liquid lipids, resulting in an imperfect crystal-type of NLC. It involves mixing spatially different lipids, consisting of many fatty acids, which introduce imperfections in the order of the crystals. The drug loading can further increase the molecular interactions, resulting in a smaller nanoparticle size.

### 2.3. Encapsulation Efficiency and Loading Capacity

The encapsulation efficiency (EE%) and loading capacity (LC%) were determined as qualitative and quantitative parameters of the production of process. EE and LC were calculated for CBD-cSLN and CBD-cNLC formulations to estimate how much CBD was inside each system. The results are shown in Table 3. The influence of varying lipid types (Compritol^®^ 888 ATO and Miglyol^®^ 812) on the more efficient loading CBD was analyzed. Consequently, CBD-cSLN achieved the EE of 74.23%, CBD-cNLC gained 70.44%, and the LC was 15.77% and 15.11%, respectively.

### 2.4. Rheology Study

The viscosity of the lipid nanoparticle-based formulations can vary from liquids to semi-solid systems. It also governs the selection of the ideal pharmaceutical formulation for the intended administration route. When nanodispersions are applied to the skin, they should show a thixotropic pattern with a high viscosity. Therefore, they can adhere and be retained on the skin site. On the other hand, in oral formulations, lipid nanoparticle-based dispersions can be used as granulating media in tablet production to convert the powder into a viscoelastic mass and then into dry-looking granules or load soft and hard gelatin capsules (either as liquids or after spray/freeze-dried).

To study the rheological behavior of the dispersions, an oscillation frequency sweep test was run for each formulation (CBD-cSLN, CBD-cNLC) at 22 ± 0.5 °C one day after production, over the frequency range of 0–10 Hz. For comparison purposes between CBD-cSLN (Figure 4, left) and CBD-cNLC (Figure 4, right), we recorded the storage modulus (G’), loss modulus (G”), and shear viscosity. The loss modulus was higher than the storage modulus over the applied frequency range for both lipid nanoparticle-based formulations. The storage modulus (viscous component) resembles how much energy the dispersion requires to be distorted, whereas the loss modulus (elastic component) translates the energy lost during the strain. At low frequencies, the shear rate is also low, which means that the capacity of the dispersions to maintain their original media strength is high. With the increase in the frequency range, the shear rate also increases, thus requiring more energy and consequently increasing the viscous component (G’, storage modulus).

The results showed that G” was always higher than G’ for both tested samples. It means that they were more elastic than viscous. When the stress was removed, the dispersions were restored to equilibrium but did not follow the same conformational path. Moreover, the dispersions did not maintain their shape, and this behavior is typical of a liquid-like character system.

In turn, the rise in the frequency increased the shear viscosity for CBD-cSLN (Figure 4, left), while for CBD-cNLC, the reverse was observed (Figure 4, right). The addition of an oily component (Miglyol^®^ 812) to the solid matrix of SLN (to obtain NLC) could result in more viscous dispersions, as seen with the higher shear viscosity values recorded for CBD-cNLC when compared to CBD-cSLN.

### 2.5. DSC

The thermodynamic stability of cSLN and cNLC is mainly governed by the lipid modifications encountered upon loading with the active ingredient. The selected solid lipid was Compritol (i.e., glyceryl behenate), a mixture of mono-, di- and tri-behenate of glycerol [36]. It is reported that Compritol has a typical crystalline structure of tryglycerols, containing tiny amounts of α-form [37]. The crystallinity index, which measures the amount of solid content inside the particles, shows the slightly solid character of SLN compared to NLC, as documented in the literature [38].

The production of SLN and NLC loading CBD resulted in the decrease in the melting peak down to 71.0 °C and 63.6 °C, respectively, with a significant depression of the crystallization index when we compare CBD-cSLN (90.41%) with CBD-cNLC (40.18%) (Table 4). These results are aligned with the loading capacity and encapsulation efficiency encountered for these particles.

Figure 5 shows bulk Compritol melted at 75.8 °C, with a recording melting enthalpy of 109.7 J/g. The bulk CBD analysis showed the active ingredient’s low melting peak at 37.5 °C (Figure 6). It has been loaded into lipid nanoparticles, contributing to the crystallinity index recorded for both types of particles.

### 2.6. SEM

The loading of CBD into Compritol-composed SLN and NLC did not compromise the typical round and smooth surface of nanoparticles, as confirmed by SEM analyses. Figure 7 and Figure 8 depict the images recorded for both developed formulations at different resolutions. These results corroborate the solid character of both types of particles described in Table 5 and the small mean particle size and polydispersity index shown in Table 2 and Table 3. These features are responsible for the typical rheological behavior of monodispersed nanoparticle dispersions.

## 3. Materials and Methods

### 3.1. Plant Extract Preparation

A hemp extract was prepared using panicles of Hungarian monoecious hemp variety KC Dora (extract B) and Polish monoecious hemp variety Tygra. Hemp was cultivated in experimental plots (Institute of Natural Fibres & Medicinal Plants). Cultivation conditions included nitrogen fertilization (30 kg/ha) and seed sowing density (30 kg of seeds/ha). Panicles were harvested at the late flowering stage when the highest content of cannabinoid compounds in dried plant material was observed (CBD—1.9153% and THC—0.0681%). The hemp extract was prepared in two-phase solvent extraction. The plant material was exposed to an organic solvent at 30 °C, and the resulting extract was concentrated in an evaporator (50 mbar vacuum). Subsequently, the extract was dissolved in ethanol and water at 80 °C. After ethanol evaporation, the extract was concentrated again (50 mbar, vacuum). The final stage of the process was decarboxylation (temperature 130 °C), which resulted in the extract containing 215.2 mg/g CBD and 13.3 mg/g THC.

### 3.2. Chromatography of Plant Extract

Approximately 150 mg of the sample was weighed into a centrifuge tube and flooded with precisely 10 mL of a MeOH: THF mixture. Then, it was shaken for 30 min, and the tubes were centrifuged at 5500 rpm. Samples were diluted 10-fold into an HPLC-type vial and placed on an HPLC autosampler. All analyzes were performed on a gradient of 0.1% H_3_PO_4_ in water with CAN, observing UV light absorbance at 230 nm on a chromatographic column with o stationary phase C18. Identification and calculation of concentration for cannabinoids were performed based on the chromatograms of the standards, from which the calibration curves were plotted.

### 3.3. Production of SLN and NLC

SLN and NLC were produced by hot high-pressure homogenization as described by Doktorovova et al. [39] with slight modification and based on preliminary studies as described by Souto et al. [40]. Briefly, the melted lipid phase (solid lipid in case of SLN; solid lipid and liquid lipid in case of NLC) was dispersed in the aqueous surfactant solution heated at the same temperature using a probe sonication (Sonics Vibracell, Newtown, CT, USA) for 15 min and 70% amplitude. A hot emulsion was obtained, then left to cool down at room temperature. For the loading of CBD, the extract was added to the inner oil phase before emulsification. The composition of the developed batches is shown in Table 5.

### 3.4. Formulation Characterization

#### 3.4.1. Particle Size, Polydispersity, and Zeta Potential

In this work, dynamic light scattering (DLS) analysis (NanoBrook Omni, Brookhaven Instruments, Holtsville, NY, USA) was used to record the mean particle size (Z-Ave, polydispersity index (PDI), and zeta potential (ZP). The samples were diluted in MilliQ water 1:10 *v*/*v* and analyzed at 20 °C, with a refractive index of 1.331 and a dielectric constant of 80.37. All of Z-Ave, PDI, and ZP measurements were recorded in triplicate.

#### 3.4.2. Encapsulation Efficiency and Loading Capacity

A volume of 5 mL of the obtained particles was submitted at centrifugation by Amicon^®^ Ultra Centrifugal Filters Ultracel (Millipore, Germany) for 25 min at 4500 rpm to isolate the particles from the suspension. This analysis aimed to estimate the CBD encapsulation efficiency. The supernatant was quantified against a calibration curve (y = 3.5316x + 0.0043, R^2^ = 0.9980) to determine the drug concentration. The calibration curve standards were prepared by diluting the CBD in acetone to ensure total dilution. The concentration of each standard of the calibration curve and the filtrate concentration were measured in a BioTek Synergy HT plate reader (BioTek Instruments, Winooski, VT, USA) at 327 nm. The encapsulation efficiency (EE) and loading capacity (LC) of CBD loaded SLN/NLC were calculated as follows [41]:$$\mathrm{EE}\%=\frac{W_{\mathrm{CBD}}-W_{S}}{W_{\mathrm{CBD}}}\times 100$$
$$\mathrm{LC}\%=\frac{W_{\mathrm{CBD}}-W_{S}}{W_{\mathrm{CBD}}-W_{S}+W_{L}}\times 100$$
where W_CBD_ is the mass of CBD used to produce the loaded nanoemulsions, Ws is the mass of CBD quantified in the supernatant, and W_L_ is the weight of lipid added in the formulation. Centrifugal filter units (µL) were used with a cut-off of 50 kDa, i.e., 50,000 nominal molecular weight limits (NMWL).

#### 3.4.3. Rheology

The rheology studies were conducted on a Malvern Kinexus rheometer (Malvern Instruments, UK). The oscillation frequency sweep test was applied over a frequency range from 0 to 10 Hz. The storage modulus (G’), loss modulus (G’’), and the complex viscosity (η*) of lipid nanoparticles were described as a function of the frequency at a constant stress amplitude of 5 Pa (linear viscoelastic region). All measurements were carried out directly at room temperature (25 °C).

#### 3.4.4. Differential Scanning Calorimetry (DSC)

This study used a differential scanning calorimeter, DSC 200 F3 Maia Differential Scanning Calorimeter from NETZSCH Premier Technologies. The liquid nitrogen’s temperature range extends from −150 °C to 600 °C. The heat flux sensor of this equipment allows high stability, improved resolution, and fast response time. Laser-guided welding processes for the sensor disk and thermocouple wires yield high sensitivity and robustness. In this work, the difference between the temperature of the samples and the standard was recorded. The samples of CBD-loaded SLN/NLC were scanned from 25 to 90 °C/min. The crystallization index (CI, %) was determined using the following equation [38]:$$\mathrm{CI}\left(\%\right)=\frac{∆H_{\mathrm{SLN}\;\mathrm{or}\;\mathrm{NLC}\;\mathrm{aqueousdispersion}}}{∆H_{\mathrm{bulkmaterial}}\times {\mathrm{Concentration}}_{\mathrm{lipid}\;\mathrm{phase}}}\times 100$$
where ΔH is the molar melting enthalpy given by J/g, the percentage of the lipid phase gives the concentration.

#### 3.4.5. Scanning Electron Microscopy

The scanning electron microscopy (SEM) analysis was performed in a high-resolution Scanning Electron Microscope (JEOL JSM-6390 Scanning Electron Microscope (SEM) at an accelerating voltage of 20 kV, Tokyo, Japan). The CBD-loaded SLN and NLC were coated with platinum (20 nm thick) using an Ion Sputter (JFC-1100, JEOL Ltd.) (Tokyo, Japan) for 5 min at 20 mA. The magnification was set at 10,000.

## 4. Conclusions

In the present work, cannabis extract was successfully loaded into two types of lipid nanoparticles, namely SLN and NLC. While the loading capacity of the extract in both systems remained around 15%, the EE was slightly higher for SLN (ca. 75%) than for NLC (ca. 70%). On the other hand, both SLN and NLC showed interesting crystallinity indices, anticipating the reason for the relatively high encapsulation efficiencies. The lipid nanoparticle dispersions followed a typical viscoelastic behavior of liquids, so further technological processing will be needed to develop a final pharmaceutical product.

## Figures and Tables

**Figure 1 molecules-28-02875-f001:**
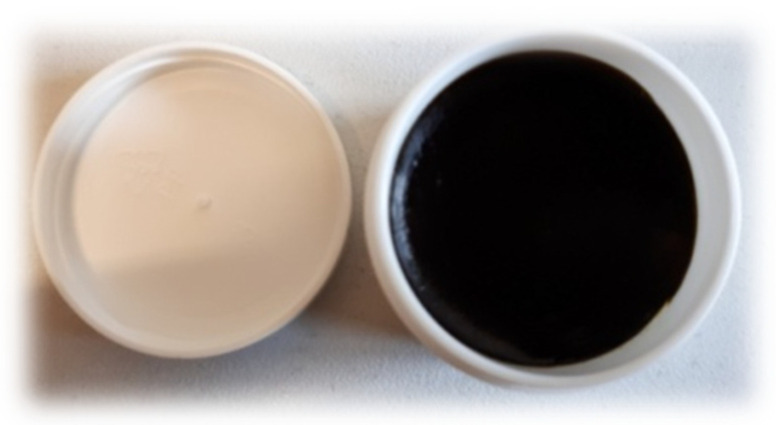
Hemp extract (*Cannabis sativa* L.) [own photography].

**Figure 2 molecules-28-02875-f002:**
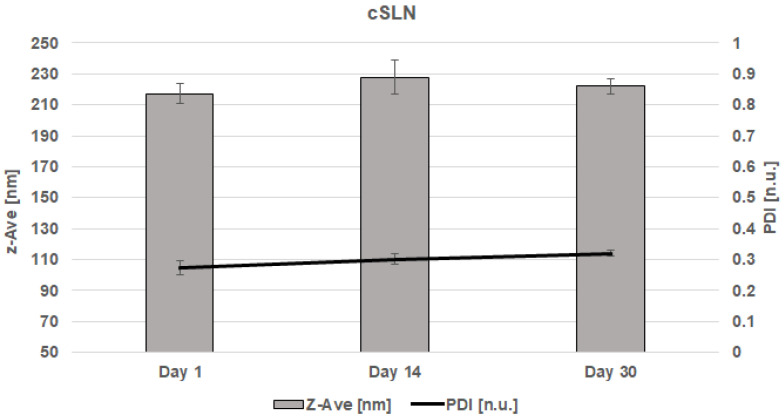
Z-Ave and PDI for CBD-cSLN on the 1, 14, and 30 days after the production.

**Figure 3 molecules-28-02875-f003:**
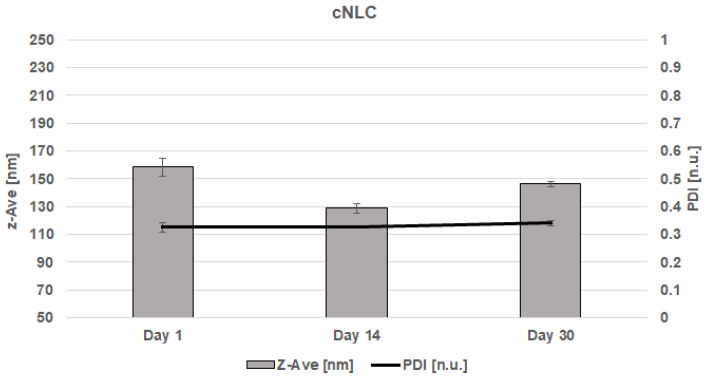
Z-Ave and PDI for CBD-cNLC on the 1, 14, and 30 days after the production.

**Figure 4 molecules-28-02875-f004:**
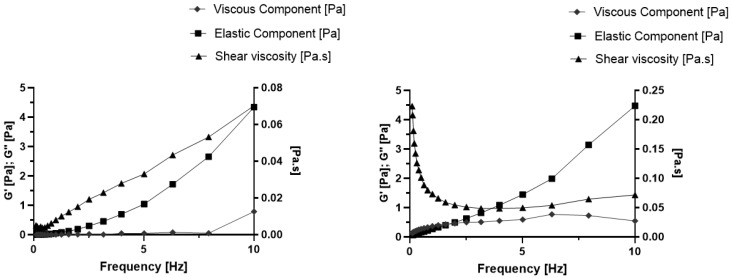
Rheological behavior of CBD-cSLN (**left-hand**) and CBD-cNLC (**right-hand**) over a frequency range of 0–10 Hz.

**Figure 5 molecules-28-02875-f005:**
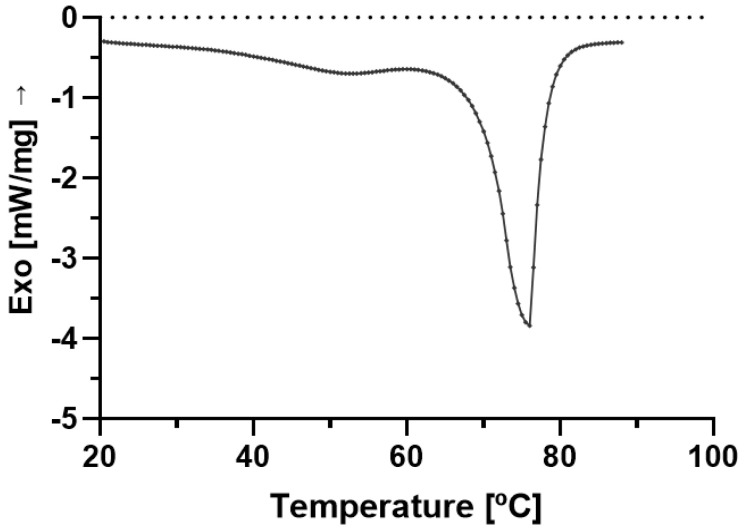
Differential scanning calorimetric profile of bulk Compritol, depicting a melting peak at 75.8 °C, with an onset temperature of 61.8 °C and end set at 84.4 °C.

**Figure 6 molecules-28-02875-f006:**
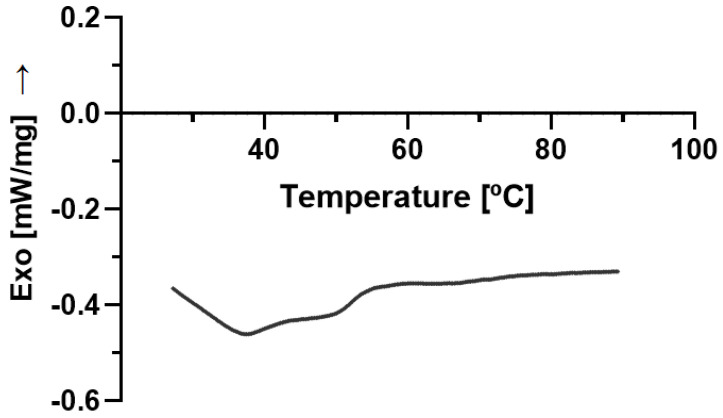
Differential scanning calorimetric profile of bulk CBD, depicting a melting peak at 37.5 °C.

**Figure 7 molecules-28-02875-f007:**
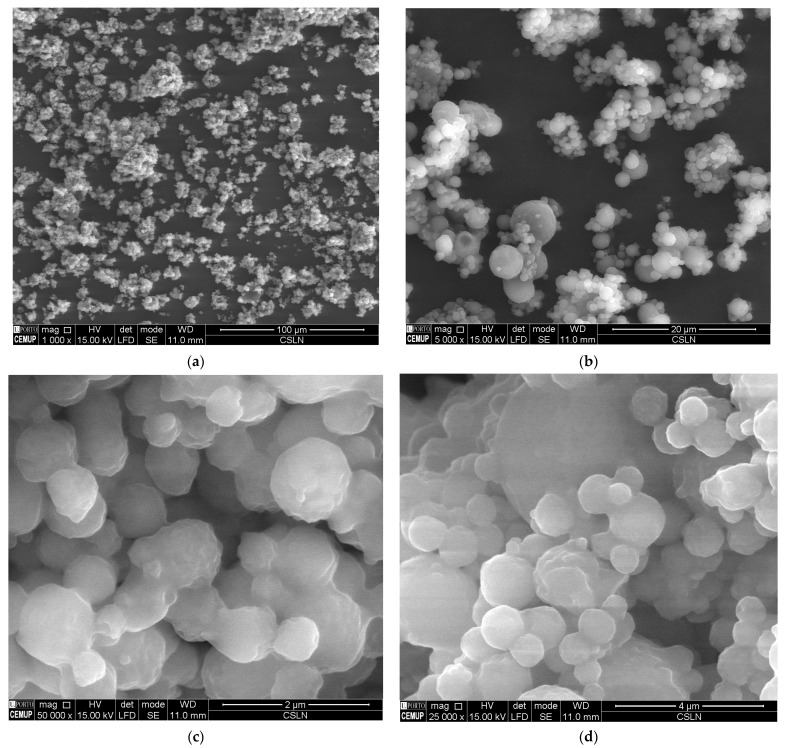
Scanning electron microscopy analysis of CBD-loaded cSLN at different resolutions, (**a**) 1000×, (**b**) 5000×, (**c**) 25,000×, and (**d**) 50,000×.

**Figure 8 molecules-28-02875-f008:**
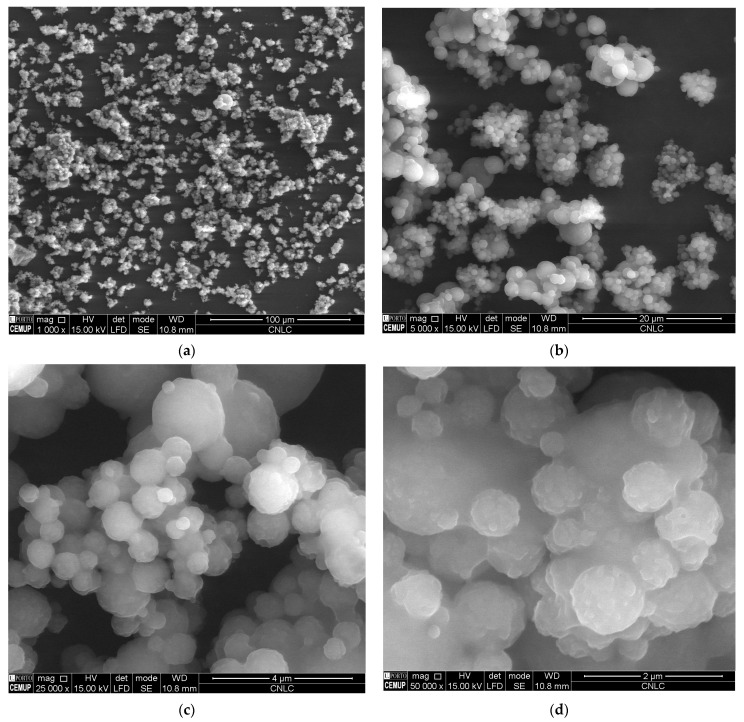
Scanning electron microscopy analysis of CBD-loaded cNLC at different resolutions, (**a**) 1000×, (**b**) 5000×, (**c**) 25,000×, and (**d**) 50,000×.

**Table 1 molecules-28-02875-t001:** Percentage of individual cannabinoids in the cannabis samples (based on Appendix A).

Lp.	1.	2.	3.	4.	5.	6.	7.	8.	9.	10.	11.
Cannabinoid *	CBDV [%]	CBDA [%]	CBGA [%]	CBG [%]	CBD [%]	CBN [%]	Δ^9^-THC [%]	CBNA [%]	CBC [%]	THCA [%]	CBCA [%]
Sample 1	0.07	7.41	0.09	0.06	3.07	0.08	0.25	0.04	0.22	0.38	0.39
Sample 2	0.06	7.09	0.04	0.06	2.97	0.08	0.25	0.04	0.22	0.37	0.38

* Abbreviations: CBDV—cannabidivarin; CBDA—cannabidiol or cannabidiol acid; CBGA—cannabigerol or cannabigerol acid; CBG—cannabigerol; CBD—cannabidiol; CBN—cannabinol; Δ^9^-THC—delta-9-tetrahydrocannabinol; CBNA—cannabinol or cannabinoic acid; CBC—cannabichromene; THCA—tetrahydrocannabinolic acid; CBCA—cannabichromene acid.

**Table 2 molecules-28-02875-t002:** ZP for CBD-cSLN and CBD-cNLC on the 1 and 30 days after the production.

Sample Name	Measurement Time (Days after Production)	ZP [mV] ± SD
empty-cSLN	0	−18.09 ± 0.81
CBD-cSLN	1	−12.99 ± 0.90
	30	−12.42 ± 2.14
empty-cNLC	0	−13.63 ± 0.89
CBD-cNLC	1	−5.51 ± 1.58
	30	−12.06 ± 1.50

Abbreviations: SD—Standard Deviation.

**Table 3 molecules-28-02875-t003:** Encapsulation efficiency (EE%) and loading capacity (LC%) for CBD-cSLN, CBD-cNLC.

	CBD-cSLN	CBD-cNLC
EE [%]	74.23	70.44
LC [%]	15.77	15.11

**Table 4 molecules-28-02875-t004:** DSC parameters for CBD-cSLN and CBD-cNLC.

DSC Parameters	CBD-cSLN	CBD-cNLC
Peak Maximum (°C)	71.00	63.60
Enthalpy (J/g)	3.960	1.934
Crystallinity Index (%)	90.41	40.18

**Table 5 molecules-28-02875-t005:** The lipid nanoparticle dispersions (% *w*/*w*) were composed of Compritol^®^ 888 ATO (solid lipid) to obtain CBD-cSLN and CBD-cNLC.

Formulation		
Ingredients	CBD-cSLN	CBD-cNLC
CBD	1	1
Compritol^®^ 888 ATO	4	3
Miglyol^®^ 812	-	1
Poloxamer^®^ 188	1.5	1.5
Water	93.5	93.5

Abbreviations: CBD—cannabidiol; cSLN—solid lipid nanoparticles-based Compritol 888 ATO; cNLC—nanostructured lipid carriers-based Compritol 888 ATO; Miglyol^®^ 812—medium-chain triglycerides extracted from endosperms of palm oil and coconut plants; Poloxamer^®^ 188—2-(2-propoxypropoxy) ethanol, nonionic block linear copolymer.

## Data Availability

Not applicable.

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
