# Peer review of "Phytocannabinoids: Chromatographic Screening of Cannabinoids and Loading into Lipid Nanoparticles"

_molecules, 2023, doi:10.3390/molecules28062875_

Round 1

Reviewer 1 Report

Title: Phytocannabinoids: chromatographic screening of canna-binoids and loading into lipid nanoparticles

Author(s): A. Zielińska, et al.

Recommendation: Publish with minor revisions

Reviewer comments: (Express Comments to the Author):

In the present work, cannabis extract was successfully loaded into two types of lipid nanoparticles, namely SLN and NLC. lipid nanoparticles have some significant advantages, resulting in that the investigation is intriguing work. SLN and NLC are the effective approaches to encapsulate natural extracts to increase the physicochemical stability of bioactive. The target species CBD is loaded in cSLN and cNLC. The determination of the encapsulation efficiency (EE), loading capacity (LC), particle size (Z-Ave), polydispersity index (PDI), and zeta potential were performed. The work has proven that the solid nature and form of cSLN/cNLC strengthen these particles' potential to modify the loaded CBD delivery profile for several biomedical applications.

I think this manuscript should be worthy to publish after the following minor revisions.

1.The data in Table 1 is inconsistent with the main text.

2.In the analyses of PDI, the significant digits of data are inconsistent.

3.In the determination of the zata potential, the measured days are fewer and failed to quantitatively confirm the results.

4.The Cannabis, acting as the organic molecule, is of hydroxyl group moiety. The hydrogen bonding structures are the important factors and interaction in between with polymers or solid environment. The work [Spectrochimica Acta Part A: Molecular and Biomolecular Spectroscopy 293 (2023) 122475] illustrated the hydrogen bonding interaction. I suggest that they could refer and cite the work.

5.In order to improve the paper’s quantity, the style and format of reference should be consistent.

6.In Table 2, what is the measurement time unit?

7.Why do you want to do Encapsulation efficiency and Loading capacity analyses? The authors should reinforce the explanations.

Author Response

Dear Reviewer, 

Please find the responses in the attached file. 
Thank you very much for all your suggestions. 
We appreciate your time. 

Kind regards, 
Aleksandra ZieliÅ„ska 
Corresponding author

Reviewer 2 Report

The manuscript has an interested approach, but there are a few questions:

1) Why CBD-cNLCs are smaller than empty? I would have expected to be bigger.

2) did you evaluate the release of the CBD? It is important to demonstrate the loading but also if the particles are able to release the CBD. I would suggest to quantify the release of the CBD, or to add some in vitro model (if it exists) to measure CBD release/efficacy. Maybe some fluorescence tag. 

Author Response

(The authors gave the same response as above.)

Reviewer 3 Report

1. What is the rationale for selecting Compritol 888 ATO, Miglytol 812 as lipids and Poloxamer 188 as surfactant? Authors need to evaluate other excipients also to evaluate their performance

2. What is the rationale for selecting mentioned quantities of excipients? Authors need to provide Statistical approach "Design of Experiments (DOE)" for finalizing the quantities and also to  finalize "Design Space".

3. For measurement of particle size and zeta potential, what is the rationale for selecting 1:10 v/v dilution and temperature of 20 C?

4. PDI is >0.2,  which is found to be samples are polydisperse . Please provide justification and PSD histograms. 

5. Please explain in detail how negative zeta potential was observed for both SLN and NLC formulations. 

Author Response

(The authors gave the same response as above.)

Round 2

Reviewer 2 Report

I would suggest having the release data to have a better manuscript.

Reviewer 3 Report

The revised manuscript looks good. The present version can be accepted.